# Why Do Vocational High School Students Opt for College?

**Wonseok Seo [1],* and Changhoon Lee [2]**

1   Department of Industrial, and Technology Education, Graduate School, Chungnam National University, Daejeon 34134, Republic of Korea
2   Department of Mechanical Engineering Education, College of Education, Chungnam National University, Daejeon 34134, Republic of Korea; harmony@cnu.ac.kr
*   Correspondence: mysws29@naver.com

**Abstract:** This study aims to elucidate how—and the underlying significance of their doing so—Korean vocational high school students decide to pursue university education rather than entering the workforce. Drawing on autoethnographic journals, the research employs a combination of Chang's descriptive-realistic, confessional-emotive, and analytical-interpretive writing methods to convey personal experiences, including the background of students and their motivations for opting for university. As an autoethnographer, I have encountered the societal perceptions of Korean vocational high schools, issues with school curricula, and misunderstandings surrounding employment. Over time, I chronicled the documented process of choosing university education, and this has been facilitated by conversations with the homeroom teacher; I have endeavored to elucidate the sociocultural implications of each student's experience through interpretive methods. This study's findings are anticipated to enhance the understanding of the fundamentals of career education in the realm of secondary vocational education and to offer a valuable reference for vocational high school educators on how to guide their students effectively. Furthermore, it should provide educational perspectives and fresh insights for vocational educators and researchers worldwide, thereby facilitating enhancements in career education policies and systems for vocational students.

**Keywords:** autoethnography; vocational high school; career technical education; adolescents; career transition; vocational education in the Republic of Korea

## 1. Introduction

Vocational education in the Republic of Korea starts at vocational high schools, as outlined by the Enforcement Decree of the Elementary and Secondary Education Act. The educational framework in Korea is divided into three principal stages: a six-year elementary school education, followed by a three-year middle school education—marking the initial segment of secondary education—and culminating in a three-year high school education, which constitutes the latter segment of secondary education. Korean high schools are separated into categories based on distinct criteria. Initially, they are classified as national/public or private high schools, depending on the entity responsible for their establishment. The schools are then further categorized into general education high schools, vocational high schools, and arts/sports high schools, reflecting their educational focus. Vocational high schools are specifically designed to nurture specialists in particular fields and to aid students in securing quality employment by providing a curriculum tailored to enhance their skills and potential. Specialized high schools conduct student admissions based on the unique selection criteria adopted by each institution. These schools are categorized into specialized high schools and Meister high schools, and according to the National Curriculum, students are enrolled in various educational tracks, such as agriculture, industry, commerce, and fishery & marine transportation. The primary objective of this system is to develop skilled professionals who can effectively meet the demands of the industrial workforce.

In the 1970s, the number of vocational high schools rapidly expanded to support the burgeoning heavy industries. Their numbers grew from 59 schools with 68,367 students in 1970 to 95 schools hosting 188,373 students by 1979 [1]. This growth, aligned with the broader economic policies for industrial advancement, sparked heightened public interest in vocational education, particularly with the national attention on the WorldSkills competition [2]. However, the economic downturns of the late 1970s and early 1980s, precipitated by events such as the Arab oil embargo and the Iranian Revolution, reduced demand for skilled labor such as vocational high school graduate skilled technicians in the manufacturing sector across the world. Also, the emphasis on student recruitment and skill cultivation often overshadowed considerations for graduates' future prospects and workplace treatment [1,3]. This shift marked the end of the expansionist policy for vocational high schools, which had been predicated on education driving national development.

The 1980s then witnessed a pivot toward policies favoring general education, with secondary vocational education gradually sidelined. As the economy evolved to favor white-collar jobs (office workers), societal preferences shifted away from labor-intensive skilled work [4]. The widening wage gap between high school and college graduates exacerbated this trend, reducing the appeal of vocational education and subsequent employment in skilled labor [1].

The 2000s saw a further decline in enthusiasm for manufacturing, with deepening societal aversion to vocational high schools. In this environment and in disregard of their original mission, vocational high schools came to be seen as an alternative for students unable to enter general high schools [5]. University education became heavily prioritized over employment, with vocational schools even adapting their curricula to resemble that of general high schools. Consequently, vocational high school students found themselves in a position where further education became a necessity. Although vocational education's main purpose is to enable the school-to-work transition by training skilled workers, many Korean vocational high school graduates, in fact, went on to higher education pursuing knowledge-based worker rather than entering the labor market [6]. This shift is evident in the annual statistics from the Korean Education Development Institute [7], which reveal a steady increase in university enrollment rates among vocational high school graduates, from 44.9% in 2001 to 71.1% in 2010. Subsequent to 2010, the college entrance rate steadily declined because the government established a policy goal to promote the employment of graduates from vocational schools and assist their further education after some period of employment through new 'Job-first and Diploma-later' policies. But as of 2023, it remains at 50.0% (with 31,456 out of 62,853 graduates entering college).

For the past 20 years, vocational high schools have been characterized by two distinct features: the cultivation of practical skills among students and the preparation of students for college entrance, with the latter achieved by mirroring the curriculum offered by conventional high schools [8]. However, due to confusion regarding the aim of secondary vocational education, namely the cultivation of industrial manpower, a new vocational education model called the "Korean-style Meister High Schools" was introduced in 2008 to cultivate advanced professional skills and qualifications among students, thereby promoting a pathway from pre-employment to postsecondary education. As of 2021, it remains a specialized educational model and has been implemented in only 52 schools with 6518 students. Plans call for converting an additional 470 vocational high schools to the Meister High School system, but this presents significant challenges in terms of time, financial resources, and other constraints.

Prior research [6,9] in Korea has identified several reasons why vocational high school students may opt to pursue further education: (1) students who enrolled due to underperformance in middle school may actually be qualified for college, exceeding the expectations of vocational schools; (2) top-performing vocational students can achieve grades comparable to their counterparts in general high schools; (3) students recognize that society prefers university credentials over practical skills; (4) students recognize the increasing credentialism and high valuation of educational attainment in the labor market; (5) students

may have opportunities for university admission based on high school principal recommendations; and (6) students may take advantage of the reinstatement—in response to South Korea's declining birth rate—of a university admissions screening system more favorable to vocational high school students. Studies in Korea have examined the motivations for college enrollment through narrow lenses, focusing on individual aspirations, policy shifts, and societal viewpoints, but have neglected the nuanced processes and experiences that may lead vocational students to higher education. This oversight has resulted in significant gaps in our understanding of the educational trajectories of such students and the complex interplay of how internal and external factors influence their decisions.

This study aims to enrich the discourse on vocational education by sharing personal narratives that shed light on the internal and external factors that influence student's decision to pursue higher education. By detailing the journey of a vocational high school student choosing to pursue a college education, this research seeks to evoke an emotional response from readers and to reignite interest in the career choices of vocational high school students. The guiding question of this study is: What educational and sociocultural factors influence vocational high school student's decision to pursue higher education in the Republic of Korea?

## 2. Methodology

This study was conducted by adopting an autoethnographic approach, with the aim of deeply understanding and describing my personal journey from graduating from a vocational high school to pursuing a university education. Autoethnography is a methodological lens that can bring to light the shared cultural contexts of the self and others, and the approach is rooted in personal experience [10]. The methodology involves examining autobiographical narratives within their social and cultural milieus; the researcher's subjectivity and the intersubjectivity between the researcher and participants are emphasized in the exposition of their experiences [11]. The subjective experiences discussed are significant as not only individual anecdotes and reflections but also as mirrors reflecting collective experiences and insights [12].

As an autoethnographer, from 2004 to 2007, I intensively engaged in self-recall and collected reflection materials to describe why and how, during my time in vocational high school, I chose to pursue a university education. On the one hand, I analyzed emails I exchanged with teachers, diaries, letters, photographs, memos, and social media posts to recall memories. I reviewed these materials, wrote them down as I remembered them, prepared a first draft [13], organized my perceptions and insights by time period by engaging in introspection and self-analysis [10], and made a final headnote [14]. In addition, to account for the ethical issues of other people's observations being recalled through memory playback, "relational ethics" was considered per Ellis's suggestion [15]. The "senior who dropped out of the fieldwork", the "homeroom teacher who counseled", and the "friend who encouraged me to join the company", all of whom are mentioned in this study, agreed that the experience was not ethically problematic after a long time, and they agreed to freely utilize the experience because they believed that sharing joint experiences could bring about professional and educational development.

For data analysis and interpretation, I applied Saldaña's elemental method [16] and effective methods, including longitudinal coding. Longitudinal coding is suitable for qualitative research that explores the changes and developments in individuals, groups, and organizations over a long period. In the fields of anthropology and education, this coding method is useful for analyzing the directionality and pattern processes of an autoethnographer's life experiences over time [16]. Although there are no typical coding and categorization methods specifically for autoethnographic data analysis, sophisticated coding and categorization of qualitative data facilitate the interpretive and writing tasks of autoethnographers [17,18]. Thorough data analysis enables valid interpretations, and valid interpretations lead to sharper analysis and writing. Moreover, analysis and interpretation

based on qualitative data yield epistemological intimacy, resulting in deeper and more valid research results [19].

The narrative style of describing my experience followed Chang's proposal [10], consisting of a mixture of descriptive-realistic writing, confessional-emotive writing, and analytical-interpretive writing to vividly depict personal experiences, reveal emotions and emotional confusion, and connect personal experiences and stories to macrolevel sociocultural discourse. Descriptive-realistic writing seeks to depict an accurate story through extensive details that create a picture for the reader. In the confessional-emotive style, the autoethnographer provides self-disclosure of the messy, emotional aspects of their experience. This vulnerability invites the reader to make emotional connections to the story. Analytical-interpretive writing includes some explanation during the storytelling of how the story pertains to the broader context [10].

To ensure the research's validity and credibility, I employed three criteria deemed relevant to articulating my experiences, per Maxwell's suggestions [20]. Verification procedures in qualitative research serve as a roadmap for maintaining the rigor of the study [21]. Initially, through "long-term involvement and rich data utilization", I recounted personal experiences and reflections concerning university enrollment as a vocational high school graduate, leveraging a wealth of data collected since 2004, including photographs, social media memos, emails, and diaries, among other materials. Second, for member checking, I circulated, via email, drafts to the relevant individuals, including three teachers, a family member, and two colleagues cited in the research. This strategy was aimed at ensuring objectivity through the incorporation of their feedback and preventing biased descriptions of each case. Last, for professional review of the study's academic contributions, validity, and so forth, I sought external validation by sharing drafts with a Ph.D. in education who specializes in qualitative research. This validation process aligns with Lincoln and Guba's criteria [21] for ensuring reliability.

## 3. Results

### 3.1. Navigating Grief and Future Uncertainties: Reasons for Enrolling in a Vocational High School

I experienced the sudden death of my father during my third year of middle school. At that time, the adults around me, including my homeroom teacher, could only offer vague words of encouragement whenever they saw me sinking into lethargy and despair. Although I could not figure out how to proceed or how to muster the strength to regain my composure, the necessity of assuming the role of breadwinner for my remaining family members was very clear to me. I could not let economic instability lead to future uncertainty. Adolescents who experience family separation due to death can feel burdened to protect grieving family members and behave maturely at home [22].

> *"When you live with purpose, you shine. You need to shine so that your father in heaven can easily find you on earth. You need to snap out of it. Do you understand?"* (Conversation with my homeroom teacher, 20 October 2003)

What exactly is a shining life? My homeroom teacher, who was also a children's literature writer, ceaselessly tried to comfort me with metaphorical expressions while intermittently offering me advice to help me overcome my despair over my father's absence. He even entrusted me to serve as "class secretary", a role responsible for recording student attendance, maintaining class operation records, and supporting the homeroom teacher's class management tasks. This was a way for me, who was drowning in despair due to my father's absence, to enhance my self-respect by performing specific duties. It was also my teacher's "secret strategy" to avoid showing favoritism toward me. One day, my homeroom teacher handed me a brochure from a vocational high school operated by a local steel company and recommended that I enroll in the school; the school, operated by a prominent local company, provided scholarships for the entire academic year upon enrollment and offered employment opportunities at the company upon graduation:

*"The reason I am recommending that you enroll in a vocational high school is the opportunities for quick employment. Of course, enrolling in a vocational high school may result in societal undervaluation, and some colleagues have concerns about the possibility of exposure to liquor or tobacco. However, I believe you can do well. It would be good for you to start working as a breadwinner first and then go to university later. Of course, you should think carefully and seriously about this before making a decision."* (Email sent by my homeroom teacher, 30 October 2003)

Because I did not have any specific career goals at the time, I began to seriously consider my homeroom teacher's suggestion of enrolling in a vocational high school. Moreover, if the curriculum involved using computers, I believed it would suit my strengths and interests to some extent and would help me find a decent job in the future.

Enrolling in a vocational high school was an opportunity to prevent future economic difficulties for my hardworking mother, and I believed that with a diligent attitude, I could take the "shortcut to a shining future." That is, vocational high schools in the Republic of Korea have served as institutions that provide technical education to prepare students for employment in industrial fields [23] and offer vocational preparatory education tailored to students' family circumstances [24]. I completed the application to enroll in a vocational high school and received a letter of acceptance in November of 2003.

### 3.2. The Stigma of "Blue Mold": Discrimination and Suppression Faced by Vocational High School Students

After enrolling in vocational high school in March 2004, I experienced what is known as entry shock [25] as I began to perceive the societal evaluation of vocational high schools firsthand. Since the mid-1980s, as secondary education became universalized, vocational high schools began to be considered a secondary option for students who did not enter general high schools [1]. The prevalent negative conception was that vocational high schools admitted students with inadequate academic performance and who were from disadvantaged family backgrounds. In an earlier period, vocational high schools in the Republic of Korea experienced rapid growth under national sponsorship motivated by a need to produce advanced technical personnel [26], but the situation had changed dramatically. Korean middle school graduates were avoiding enrolling in vocational high schools, resulting in many schools struggling to secure enrollment quotas. The perception of vocational high schools as a fallback option for those who could not enter general high schools intensified the popular aversion to them.

For example, in 2007, D Vocational High School in Korea was opposed on NIMBY (not in my backyard) grounds by residents. It eventually faced closure after residents associated the study body with criminality and felt the school threatened the safety of nearby residents and would negatively impact the local children [27]. Vocational high school students were labeled as "lazy" and "losers", terms that would stigmatize me when wearing the school uniform. Walking down the street, people murmured about "blue mold" (a derogatory term referring to students wearing blue uniforms) wandering around without studying, and the exasperated sighs and cold stares from adults were concordant with not just my own feelings of inferiority but the reality of vocational high school students in general at that time. One day, suffering from inexplicable discomfort and abdominal pain before the first mid-term exams, I could not contain my body and visited a small clinic in the neighborhood with my mother, where I caught a glimpse of the societal evaluation of vocational high school students.

*Doctor: What brings you here?*

*Me: I feel like I'm going to vomit continuously, and my stomach hurts so much that it's hard to move.*

*Doctor: (after examination) It seems like you've been under a lot of stress while studying. You seem to have been very tense. Your mother should take good care of you. Which school do you attend?*

*Mother: Just a nearby school.*

*Doctor: Is it the place nearby called C Vocational High School? That school used to be well-known, right? But now, I don't think there would be much stress from studying there.* (Memo, June 2004)

Regardless of how I lived my life, attending vocational high school would be a stigmatizing experience and the basis for judgment from outsiders; it would be a symbol of incompetence. Talking about my school was a negative experience that made me feel ashamed and resentful. I think that my abdominal pain and discomfort were a manifestation of the burden of taking on the role of breadwinner in my father's absence and the shame and stigma associated with attending a vocational high school, which caused my heart to cry out in resistance. A difficult-to-describe bitterness began to grow in the recesses of my heart. Although a health professional diagnosed the symptoms as stress-related, my interpretation is that this stress, and its respective symptoms, were caused by social pressures and my personal experiences.

### 3.3. Voice of Resistance and Critique: Growth of Vocational High School Students

Vocational high school students showed more interest in activities that could immediately earn them money rather than developing their individual abilities and qualities through classroom learning and repetitive functional training. Graduating and finding a job existed somewhere in the obscure future, and they searched for part-time jobs that could provide immediate income. Students focused on their economic livelihood, perceiving that learning would not immediately benefit their lives. This was because they did not receive career guidance according to their aptitude, abilities, and interests but rather attended the school due to difficult family circumstances and low academic performance, which led them to struggle to create their lives independently. The fact that over half of specialized high school students participate in part-time work each year [28] supports this notion.

*"Studying to go to college doesn't seem to suit me anyway, and I just want to live my life my way. To be honest, coming to vocational high school is proof that I didn't study well in middle school. I'm not confident, and my life is not fun. It seems like earning money by working is a better choice than studying now. I don't even know whether what I learn at school would really help. I feel sorry for asking for pocket money at home. But working helps build experience. It's a bit tough working part-time now, but I'll manage somehow. Anyway, it's better than studying. Let's go work together."* (Conversation with a friend, April 2004)

Even students without special backgrounds found that the easiest entry to work was by taking simple part-time jobs such as "parcel loading and unloading" and "convenience store jobs" after school. Working part-time was both a pleasure and a survival strategy, allowing them to design and pursue their lives independently. Therefore, Korean specialized high school students tended to decide on their careers early and to develop career awareness [29,30], resulting in relatively fast employment. When considering that the fundamental cause of Korean students' unhappiness with school life or life in general is their inability to design and pursue their lives independently [31], such behavior can, in contrast, be seen as evidence of an active attitude toward life.

However, classrooms dominated by "working students" transformed into "sleeping room" for recovering mental and physical energy rather than serving their original purpose as a site of instruction. Some teachers implicitly accepted this atmosphere, neglecting students by unilaterally deciding to teach or not teach during self-study hours. The premise that teacher expertise is maintained and improved by not lowering educational expectations for students and taking appropriate measures to reach each student [32] contrasts with this situation of classrooms where teachers lack enthusiasm, no one wakes up sleeping students, and no one feels any guilt about sleeping during class. Thus, I witnessed the gradual collapse of the school-level curriculum and operating system firsthand. One day before summer vacation, I witnessed a senior student returning from a placement who was

being scolded for finding the company too difficult to handle. Placement is a system where vocational high school students gain practical experience before formal employment at companies recommended by the school.

> *Teacher: This is the company I recommended—how can you act so irresponsibly there?*
>
> *Student: Teacher, I was only asked to carry heavy things every day and do simple assembly work. They scolded me if I took a short break because of the difficulty of carrying things. How can I endure this? I thought I was going to operate a milling machine, but it's different from how you explained at first.*
>
> *Teacher: If you had shown a little diligence, you would have been given more important tasks and machining work there. Which company entrusts important tasks to someone from the start? If you had gone to a good company with your current skills, you should have endured it well. Do you know how much trouble the teachers went through to get you employed? If you can't handle this, how will you survive in society later? Because of you, our school won't be able to recommend students to that company anymore.* (Diaries, July 2004)

As a result of this incident, the teacher felt that the good relationship they had cultivated with the company had been irreparably harmed, and consequently, other students would be deprived of the opportunity to work in that company. Further, the teacher branded the student as "irresponsible", gossiping about him widely and depriving him of the opportunity to get a job at another company. Is it inevitable to experience irrational and discriminatory practices during one's first social experience of a job placement? Why must we endure high-intensity labor environments unrelated to our majors and accumulate career experience? Some career guidance teachers, as part of placement, often dispatched students to industrial sites without considering their individual majors, aptitudes, and interests [33,34]. They frequently prioritized the size of the company and arranged placements in places where cooperation was easy. It became customary at that time for companies to request interns not for educational motives but to fill vacancies for simple duties and tasks [35]. Consequently, the quality of employment varied depending on which company happened to be hiring that year.

Meanwhile, regardless of major or field of recruitment, entering a large corporation as a production worker or government as a technical employee would receive applause, and the employment results would be celebrated with banners hung at the school gate to boost the school's educational achievements. School banners were deployed to highlight educational achievements, enhance the image of the school in the region, and highlight the identity of being a school that guides students to successful employment. However, this practice fails to reflect the respect or value attributed by the surrounding community to students who succeed in employment after graduation. This discrepancy can be understood within the context of the diverse values regarding education and employment in Korean society, influenced by social class structures. Education is often perceived not only as a means of acquiring knowledge but also as a pathway to economic success. Consequently, the educational pursuit may become overshadowed by the focus on employment prospects. Thus, the paradox between the celebration of educational achievements within school culture and the lack of recognition from the surrounding community reflects the complex nature of the education and employment systems in Korean society.

Student employment guidance was a crucial responsibility of vocational high school teachers [36], and the school's employment rate was a significant "root indicator" because it was necessary for regional education offices to secure funding related to vocational education [35]. Observing distorted teaching practices and educational operations, I began to form a sense of resistance and critical opinions toward vocational education sites, and I started to contemplate what educational philosophy, responsibilities, and mission vocational education teachers should embody.

### 3.4. Warm Encouragement and Guidance from Teachers for Career Decision-Making

During my ongoing reflections on the role of teachers, I happened to come across a publication by a local educational authority titled "Success Stories of Vocational High School Graduates." It was a collection of essays featuring individuals who, as vocational high school graduates, ventured into new fields rather than simply entering the workforce. Through these stories, I realized that as a vocational high school student, there was potential to explore new career paths beyond traditional employment after graduation. The idea that I could contribute to changing the negative image associated with vocational high schools appealed to me. Sharing my thoughts with my homeroom teacher at the time, I expressed concerns:

*"I was a vocational high school graduate too, did you know? My family was really struggling financially. So, I understand your feelings better than anyone. I've faced many hardships in my life too. We were always struggling to make ends meet. Since there was no one around to talk about these things, I just kept thinking on my own. That's why I entered vocational high school, because I thought I could earn money quickly. But it turned out it wasn't as easy as I thought. If I had made fewer mistakes, it would have been better. How about becoming a vocational high school teacher after graduating from C University? You have good grades, so let's focus on preparing you for the college entrance exam right now. Your diligence is your great strength. If you don't give up, doors will surely open for you. I've always lived with that belief. But when you graduate from vocational high school, don't give up on your major in mechanical engineering. Someday, you'll surely find a use for it. If you have the opportunity, it would be good to study Chinese or Japanese, languages spoken in neighboring countries."* (Conversation with the teacher, December 2004)

The teacher maintained a warm and encouraging demeanor. Surprisingly, he revealed that he was also a vocational high school graduate, which elicited a sense of psychological empathy from me. He strongly recommended that I become a vocational high school teacher, which fascinated me. I thought to myself, "*He's been through the same situation as me.*" Feeling a sense of psychological affinity, I also began to think that I should become a teacher and serve as a role model. Becoming a vocational high school teacher seemed like a reasonable plan to me, offering a route to recognition and love from others. It was also a strategy to compensate for my wounded self-esteem by presenting myself as someone with greater abilities than I had.

*"It's so attractive to think about changing and teaching someone. The time spent talking with the after-school teacher is truly enjoyable. Time flies by. Anyway, becoming a teacher seems like a good opportunity to change the negative image of vocational high school students that is so pervasive in society. I also want to change the view that vocational high schools are attended only by those who lack abilities. Becoming a vocational school teacher seems like a decent thing to do."* (Personal memo, 23 December 2004)

The teacher advised me that upon entering university, I should take a course load based on mathematical knowledge and scientific reasoning, with some English language study as well, to establish a solid foundation. At the time, the education program at C University, which trained vocational high school teachers, required the completion of 20 credits of subject matter education and 120 credits of subject content education, mainly comprising engineering-related courses, which naturally depended on having built a strong foundation during high school. Aware of my difficult financial situation, my homeroom teacher assured me that a lack of money should not interfere with my studies. He personally procured various textbooks and problem sets I needed for studying, and in after-school sessions, he explained basic physics concepts to me. In fact, these textbooks and problem sets were "teacher's guides" provided by publishers for teaching research purposes. They were filled with detailed explanations and teaching methods for each topic, with answers provided for every practice problem. Although it was tedious and bothersome to erase or

cover the answers every time I worked through problems, it was a much lighter burden than asking my mother for extra money to purchase study materials.

The vocational high school–level curriculum was structured around specialized subjects rather than an education program designed for university entrance. Working through separate textbooks provided by my teacher and solving problems was not just basic preparation for taking the mathematics proficiency exam required for university admission; it was also necessary pre-requisite learning to prepare for potential difficulties in university-level studies. A significant number of vocational high school graduates had reported struggling after entering university due to inadequate foundational knowledge. Taking advantage of this situation, private academies near vocational high schools in the Republic of Korea catered to students aiming to attend higher education by offering lessons in advanced subjects not included in the school curriculum, giving students the opportunity to hone the skills and foundations necessary for taking the university entrance exam. Completing schoolwork could result in the attainment of the minimum grades and attendance necessary for university admission, but some students sought additional preparation for university through these academies. It was a curious phenomenon that despite enrolling in vocational high schools due to economic difficulties, some students ended up investing in private academies for further education.

Encouraged by the guidance and sincerity of the teacher who encouraged me to become a vocational high school teacher, I evolved into a "unique vocational high school student" preparing for university entrance rather than just a "job seeker." Within the vocational high school society, I embraced an unconventional, proactive attitude, gradually becoming an outlier.

*I'm very proud of you, as a teacher, seeing you mature day by day. I think it was around the middle of March of your freshman year when I first saw you, and I thought you would be a very good student. You have grown up wonderfully, meeting your teacher's expectations. I hope that your dreams will come true when you graduate. I will pray for you.* (31 December 2005, email from the teacher)

I incorporated various advice and encouragement from those around me into my daily life. I arrived at school at 6:00 a.m. to memorize English vocabulary and entire English sentences. After school ended at 4:00 p.m., I stayed in the workshop until 8:00 p.m., training in the skills necessary to obtain a qualification in mechanical drafting. Once the day's work was done, I went to the nearby library to prepare for the college entrance exam, studying until late at night. These efforts led to increased confidence in my studies and improvement in my scores on practice college entrance exams. This positive learning experience served as a solid foundation for my attempts to enter university. Immediately after graduating from the undergraduate program in 2007, I entered C University's Vocational Education and Training major as a national scholarship student, receiving a full scholarship for four years and living expenses. Subsequently, I went to graduate school and am currently pursuing my life journey as a researcher in the field of vocational education and training in the Republic of Korea.

## 4. Conclusions and Discussion

This research was a deeply personal but universally resonant exploration into the motivations of Korean vocational high school students to pursue university education. I adopted the lens of autoethnography, and in this study, I have navigated the complex sociocultural landscapes that shape vocational students' decisions and illuminated the pivotal moments of resistance, critique, and eventual transformation that I experienced as a representative of this demographic.

The journey chronicled in this paper highlights the confrontation with societal biases against secondary vocational education in the Republic of Korea, shedding light on the systemic undervaluation of vocational pathways and their perceived inferiority to academic routes. This study's narrative vividly illustrates the stigmatization faced by vocational students and the profound impact of such perceptions on their identity, self-esteem, and

career choices. For me, these led me to feel I lived a fragmented existence, being unable to belong anywhere, and I experienced feelings of emptiness and depression, leading to narcissistic injury [37].

This research also underscores the significance of teacher guidance and the transformative power of mentorship in navigating educational and vocational crossroads. The encouragement and counsel of educators, especially those who have had similar experiences to their students, represent a beacon of hope and direction, profoundly influencing the decision to pursue higher education. I derived the psychological energy to face and overcome the challenges, pains, and frustrations that I inevitably encountered in my life, as well as grace and composure, through conversations with my teacher. The choice to enter university stemmed from trust, respect, and human interaction with the teacher [38,39]. Through the security, care, and assurance provided by the teacher, I realized that I could do something and that I was a valuable individual and a responsible person, and I gradually healed from my wounds.

This narrative not only speaks to the personal growth and aspirations of the author but also serves as a testament to the potential of empathetic and informed guidance in shaping students' futures. Furthermore, the study captures the resilience and agency of vocational students as they seek to redefine their paths and aspirations beyond the constraints of societal expectations and limitations. The author's transition from a vocational high school to university and eventually into academia challenges the conventional narratives surrounding vocational education, suggesting a reconsideration and broader recognition of its value and potential.

This autoethnographic account calls for a critical reevaluation of vocational education policies and practices, advocating for a more inclusive and equitable educational landscape that recognizes and nurtures the diverse talents and aspirations of all students. It emphasizes the need for policies that do away with the stigma associated with vocational education, promote the value of vocational pathways, and ensure meaningful career guidance and support for vocational students.

Building on the insights gained from this study, future research should explore the experiences of a broader cohort of vocational students across different contexts and cultures to understand the universality and specificity of the challenges and opportunities they face. In addition, there is a need for longitudinal studies tracking the career trajectories of vocational students after university to assess the long-term impact of their educational choices on their professional and personal development.

In conclusion, this research not only contributes to the discourse on vocational education and career decision-making but also advocates for a more nuanced understanding and appreciation of the vocational pathways. It is a call to action for educators, policymakers, and society at large to foster a more supportive and inclusive environment for vocational students, enabling them to fully realize their potential and aspirations. The quality of education depends on teacher characteristics, especially in the case of vocational high schools, where educational success relies on teachers' professionalism. Vocational high school teachers should collect extensive information and data on career choices and provide counseling based on specific experiences to positively influence students. Therefore, I hope that the insights from this study contribute to the ongoing efforts to better understand and support the career choice process of vocational high school students. I recognize that this research is part of a broader conversation and encourage further studies to build upon these preliminary findings; articulating and establishing the unique characteristics of secondary vocational education is crucial for establishing the identity of vocational education.

**Author Contributions:** Conceptualization, W.S. and C.L.; methodology, W.S.; validation, C.L.; formal analysis, W.S.; investigation, W.S.; writing-original draft preparation, W.S.; writing-review and editing, W.S. and C.L.; supervision, C.L.; All authors have read and agreed to the published version of the manuscript.

**Funding:** This research received no external funding.

**Institutional Review Board Statement:** Ethical review and approval were waived for this study due to it not involving the collection or recording of personal identifiable information of research subjects, in accordance with the Korean Bioethics Safety Act.

**Informed Consent Statement:** Informed consent was obtained from all subjects involved in the study.

**Data Availability Statement:** The original contributions presented in the study are included in the article, further inquiries can be directed to the corresponding author.

**Conflicts of Interest:** The authors declare no conflicts of interest.

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
