# Peer review of "Why Do Vocational High School Students Opt for College?"

_education, doi:10.3390/educsci14050534_

Round 1
Reviewer 1 Report
Comments and Suggestions for Authors
· Good contextualization of the locality where the study takes place
· “ to promote heavy and chemical industries” – check the use of heavy in this context
· “Why did I choose college education as a vocational high school student, and what growth and development did I experience during this process?” – I would use another method to highlight the research question
· Very unique and daunting methodology, well supported by literature
· P. 5-6 – it is my understanding that the approach focused by the author is based on his/her personal experiences, nevertheless, on pages 5 and 6 I came across narratives that reflect either the author’s dialogue with other agents or the author’s observations of third parties. With that in mind, I would strongly suggest adding these elements to the methodology so it is clear for the reader. Also, on this matter, I would also like to suggest some notes or clarifications that the author’s observations regarding third parties will not compromise their identities for ethical matters;
· The paper throws enlightened perspective on the historical facts surrounding Northern Korean educational system which is also relevant for research lines which focus on vocational high school education around the world.
Reviewer 2 Report
Comments and Suggestions for Authors
Reading this article, I felt that it had a great potential, but it needed to be developed further, as it raises several questions that the author does not answer.
In general, Vocational High Schools are not a universally implemented teaching model and their structure may vary. Therefore, it is important the author to provide a clear explanation of what the model is applied in Korean. The lack of clarity in the text may pose a problem, particularly for readers who are not familiar with Korean culture.
At times, the author assumes that certain concepts are self-evident and does not provide sufficient explanation (for example: on the line 37, explain which mean “second oil shock”; on the line 39, explain what is “developmental education paradigm”; on the line 42, explain which means “white-collar jobs”).
On the other hand, the article needs further development of the state of the art, with references also to non-Asian authors.
In the abstract, the author states that the purpose of the study is to interpret the process and the “profound” meaning… In this case, the adjective "profound" is only relevant because the study is an autoethnography. However, at this time, the readers do not know that, and they may wonder: What does the author mean by “profound”? Thus, the language used should be more clear and objective, avoiding adjectives.
Korean vocational high school students may choose to attend university for various reasons, such as a desire to continue studying or not needing to find a job (students don't choose university as a career just because of the salary). Therefore, they do not, necessarily, need to have a "profound meaning" to choose to go to university. So, this study aims to explore the “reasons/ motivations” that lead them to pursue (or not) higher education.
It is clear that the author's intention is to provide a personal account of their experience and its significance in in his trajectory from vocational secondary education to university education. However, certain experiences seem to be more relevant to the field of psychology, as it discusses emotions, than to the field of education itself.
Another question that arises is: Why doesn't the author explain the situation of vocational secondary education in Korea today? The fact that the author experienced it in 2004, and it's now 2024 (20 years later), calls for an update on the current situation of this type of education in Korea. In 20 years many things may have changed.
In the pg 5 (lines from 213 to 217) can you prove that your abdominal pain and disconfort were a manifestation of the burden of takining on the role of breadwinnwe in you father´s absence? Did the doctor confirm this hypothesis? It is not clear. In the same way, in the pg 2 (on the line 55), when you say “Previous studies in Korea have superficially interpreted the phenomenon…”, which studies? Can you prove that they have superficially interpreted the phenomenon?
Missing references, for example, on the line 224; from 289 to 294;
Another question is: If, when students enter a large company through the school, their employment results are celebrated with banners hanging from the school gate to show the school's success, why doesn't the surrounding community respect these schools and the students who study there? It seems like a contradiction to me.
More questions: If you entered at University C after 2007 (it was 2008?) receiving a full scholarship for 4 years (until 2011?), and then you went on to postgraduate school, and you are currently building your life path as a researcher in the field of vocational education and training in the Republic of Korea, does it mean that since 2011 you have been working as a scientific researcher? Haven't you realised your dream of becoming a teacher in vocational education?

Reviewer 3 Report
Comments and Suggestions for Authors
Dear author(s),
This study is specific. The author(s) stated that primary objective of vocational education in the Republic of Korea is to cultivate skilled professionals who can effectively meet the demands of the industrial workforce.
Therefore, the author(s) detected a problem in previous studies in Korea have superficially interpreted the phenomenon of college enrollment from a policy perspective and the perceived necessity of higher education, citing vague aspirations, discrimination against high school graduates in employment, and mismatches in majors as reasons for choosing university.
The literature review includes Korea’s literature naturally, but this is critical limitation. Additional, Although the author(s) explianed that explore the process of college enrollment among vocational high school students to describe own experiences, there is a gap in the paper as regarding to reference the previous studies, which was the other critical limitation. The purpose of this study is too mixed and it includes a few focus in tems of the author(s)' own experiences, trying evoke emotional experiences among readers, and raise societal interest in vocational high school students' career choices. The general view of this section in the paper seems dispersed and weak.
This study was a qualitative research to interpret the process and profound meaning behind Korean vocational high school students' decision to opt for university instead of seeking employment on focus the author(s)' experiences.
The author(s) based on their own experiences with employing a blend of Chang's descriptive-realistic writing, confessional-emotive writing, and analytical-interpretive writing methods.
The method section seems mixed. There is no any clear diagram on how to implement the tools. It was ambiguous in terms of obtaining the data.
Regarding the findings, the author(s) expected to understand of the fundamentals of career education in focus of secondary vocational education for profound to guide the students.
Regarding the Conclusions and Discussion
This sentence is too assertive in the paper such “Therefore, the research results so far hope to serve as a basis for understanding and improving the career choice process of vocational high school students.”
However, I join with the author(s) who stated that quality of education depends on teacher variables, especially in the case of vocational high schools, where educational success relies on teacher professionalism.
Good luck
Comments on the Quality of English LanguageMinor editing of English language required
Reviewer 4 Report
Comments and Suggestions for Authors
As an autoethnographic approach from a first-person perspective, this research endeavors to facilitate the vocational high school students' career decision-making process. It places great emphasis on identifying and establishing the distinctive features of secondary vocational education in order to establish its identity. Additionally, this paper aims to instill hope in students and foster opportunities for addressing societal biases while expanding educational horizons.
Although it is very hard to find justification for others, since the differences in experiences, family matters, socials issues etc, this research suggest the important role from the teachers. A homeroom teacher who graduated from a vocational high school for providing ongoing counseling and encouragement that helped them set a goal of entering university. The teacher provided security, care, and assurance that helped the author realize their value as an individual and become responsible. Overall, this manuscript is about overcoming biases against vocational high schools through positive human interactions such as counseling, encouragement, trust, respect, security, care,and assurance; could be well accepted as considerations to reform the education policy in different areas.
Things to considers: this research conducted in 20 years back. How about the recent condition..? While information technology, economic growth, social condition and many factors have impacted us. These need to be discussed accordingly, so the finding of the research found its novelty.
Round 2
Reviewer 2 Report
Comments and Suggestions for Authors
The article has improved significantly.
However, there are still some issues that need to be addressed. I have provided specific revisions within the article. Once these changes have been made, I recommend publishing the article. Thank you.

Reviewer 3 Report
Comments and Suggestions for Authors
Dear author(s),
Regarding the previous emphasized points, it seems that literature review has still been keeping its limitation when reviewed current version, although adding a few sources. However, there is a little bit advance in the study purpose. For the research goal, this paper has two main subject and this situation need to be thought on whether this study is divided into two parts.
As the method, there is no any clear diagram about implementation. The assessed experience could be well if it is based on more experiences belonged someone with the author’s ones, if possible. However, some assertive sentences seem nice more than present ones. Consequently, I can suggest the author(s) this paper reevaluate as dividing two parts as experiences and present situation comparatively.
Good luck.
Comments on the Quality of English LanguageMinor editing of English language required
Reviewer 4 Report
Comments and Suggestions for Authors
In my personal opinion after careful consideration and analysis, I have concluded that the updated version of the manuscript provides a more accurate and significant comprehension of the situation at hand.
It is apparent that there exists a significant deviation between the advancement of technology and societal shifts in comparison to the improvement process for vocational schools. This manuscript serves as an invaluable source of evidence that highlights this pressing issue, underlining its importance and urgency, especially for the school management process and quality assurance objectives.
Congrats and all the best!
